# New End-to-End Strategy Based on DeepLabv3+ Semantic Segmentation for Human Head Detection

**DOI:** 10.3390/s21175848

**Published:** 2021-08-30

**Authors:** Mohamed Chouai, Petr Dolezel, Dominik Stursa, Zdenek Nemec

**Affiliations:** Faculty of Electrical Engineering and Informatics, University of Pardubice, 532 10 Pardubice, Czech Republic; petr.dolezel@upce.cz (P.D.); dominik.stursa@upce.cz (D.S.); zdenek.nemec@upce.cz (Z.N.)

**Keywords:** safety systems, head detection, head counting, semantic segmentation, parallel networks, DeepLabv3+

## Abstract

In the field of computer vision, object detection consists of automatically finding objects in images by giving their positions. The most common fields of application are safety systems (pedestrian detection, identification of behavior) and control systems. Another important application is head/person detection, which is the primary material for road safety, rescue, surveillance, etc. In this study, we developed a new approach based on two parallel Deeplapv3+ to improve the performance of the person detection system. For the implementation of our semantic segmentation model, a working methodology with two types of ground truths extracted from the bounding boxes given by the original ground truths was established. The approach has been implemented in our two private datasets as well as in a public dataset. To show the performance of the proposed system, a comparative analysis was carried out on two deep learning semantic segmentation state-of-art models: SegNet and U-Net. By achieving 99.14% of global accuracy, the result demonstrated that the developed strategy could be an efficient way to build a deep neural network model for semantic segmentation. This strategy can be used, not only for the detection of the human head but also be applied in several semantic segmentation applications.

## 1. Introduction

Every day, more data is provided by the sensors around us. These sensors are themselves more precise and therefore more talkative. Some of them record environmental data and translate them into a vocabulary understandable by humans (RGB camera, IR, ultrasound, telescopes, etc.). Others feed the inputs of smart systems. The field of computer vision involves processing the data provided by the many image sensors available to us. This is done in order to enable a computer to perform specific tasks without the help of humans. Despite significant advances in this field of research, the sensor-computer pair is still far from matching the performance of that formed by the eye and the brain [1]. One of the major issues in the field of computer vision is the automation of recognition and detection systems. Indeed, this field continues to develop to assist humans in these daily tasks. This has occurred particularly in recent years with the push of a new well-known artificial intelligence domain: Deep Learning.

In the field of computer vision, object detection consists of automatically finding objects in images. That is to say, giving their positions in these images. The most common applications are security or safety systems (pedestrian detection, behavior identification, etc.). Recognition is a related problem of knowing to which category of object the content of an image, or part of an image, belongs.

In crowd images, people are sometimes partially visible; we just see the upper body of the people. We need to design a more specific system to solve this problem. Several techniques and approaches exist, some try to estimate the number of people directly [2,3] and others try to go through an intermediate step by calculating a density of people in the image [4,5]. The latter allows retrieval of more information and a better interpretation of images.

In this study, development of a passenger flow monitoring system has been done. The proposed system will be able to manage different distances between the head and the sensor (overhead camera). The datasets were taken from varying distances, and further, over the stairs where the distance from the sensor and head may vary. The purpose is the detection and counting of passengers entering and leaving a means of transport (or in many other environments). This can give many benefits in daily operational life. One example is counting the exact number of passengers in areas such as public transport. This allows the forecasting of passenger flows, planning transport schedules, and monitoring the loading of transport vehicles. A second benefit is tracking people in a video, notably in security systems to obtain the path of an individual and analyze his movement. A third advantage is in stores and malls where this type of technology is used to collect data and improve business strategy.

Considering the fact that object detection is very far from being resolved (especially for overlapping instances) [6], we have assumed the use of deep learning object detector architecture, which locates the main object instances in the image. Taking into account the location of all the instances, the counting becomes trivial. Identifying, counting, and measuring the number of persons in airports, buses, trains, subways, etc., in real time, is one of the goals of this study.

Several difficulties were encountered during this study. One of the first classic difficulties is the disparity in the appearances of the heads. This difficulty is called intra-class variability [7]. Despite appearances that are sometimes very different in a given class, algorithms are expected to be able to correctly predict this class. Another classic difficulty is the high variability of background. We define any region of the image, that does not contain a head, as the background. Unlike detection in controlled environments, the backgrounds of the processed images can be diverse, and it is not possible to model them accurately without restriction. Otherwise, the acquisition conditions and the type of sensor influence the quality of the images, in which poor quality is one of the difficulties of such study. Finally, the dataset itself can create a problem that makes the models inoperative. This is due to the small size of the dataset and the preparation of the ground truth by a human who can make an error at any time.

To solve those difficulties, a new end-to-end strategy (which accepts raw images as input and directly generates a set of bounding boxes of objects as output) based on DeepLabv3+ [8] semantic segmentation has been developed. It is essential to mention that semantic segmentation has never been applied before in the field of human head detection. Not only that, the approach is based on two parallel Deeplapv3+ to improve the performance of the proposed system. We did not directly work with the bounding boxes. Instead, we used two types of ground truths extracted from those bounding boxes (each one is applied to one of the two parallel networks). The first is created by transferring the bounding boxes to masks (in elliptical format) to have binary images of the heads. The second is set up by extracting the centers of each object and building a new ground truth which consists of binary images containing the centers of each object. For this second ground truth, to avoid the problem of detecting very small objects, we chose not to have only the center, but a circular area of 10-pixels that surrounds the center of the object.

The approach has been implemented in our two private datasets as well as in the PAMELA UANDES [9] public dataset. The result demonstrated that the fusion of the two parallel networks that we made could be an efficient way to build a deep neural network model for semantic segmentation. This can be applied, not only for the detection of the human head, but also in several semantic segmentation applications.

Our innovation/substantial contribution in this study are as follows: The proposal of a new semantic segmentation strategy based on two parallel networks (we believe that it can give good results in many applications of semantic segmentation)A new specific preparation of the ground truth to improve the result of semantic segmentation has been proposed (can also be applied in many applications).An in-depth study of the effect of changing the base DCNN architecture of Deeplabv3+ on detection performance has been investigated.The application of the models proposed in 3 different datasets (two of them are private and one public)

The remainder of this article is organized as follows. In Section 2, we provide an overview of the research that has been carried out in previous years in this area. Section 3 gives a description of the different phases involved in the proposed system, as well as the background information. The benchmarking study and the evaluation matrices on which we based are presented in the Section 4. Otherwise, Section 5 shows the experimental results in which we analyze, interpret and discuss our results. Finally, Section 6 closes the article with conclusions and future work.

## 2. Related Works

Deep learning models have seen significant success in a wide range of applications in recent years. In fact, different deep learning methods have been introduced for image and motion segmentation [10,11,12,13], detection [14,15,16], tracking [17,18] and classification [19,20]. Due to the success of deep learning methods, CNN models have also been used in the literature to provide relevant information on the number of people present on the stage.

Since our framework is based on a deep learning approach, we next discuss relevant work of applying head detection through the application of artificial intelligence.

Le Thanh et al. [21] presented a comparison of face and head detectors (used on public datasets) based on CNN for video surveillance applications. They used several single-pass and region-based CNN architectures (SDD, R-CNN, Faster R-CNN, and PVANET) with base detectors. El Ahmar et al. [22] introduced a real-time approach for the detection of the human head and shoulders from RGB-D data based on image processing and deep learning. They added Candidate Head Locations (CHL) to exploit depth data to improve detection accuracy.

Introduced in [23], the authors focused on detecting human heads in natural scenes by a combination of context-aware models and the CNN-based local detector. Moreover, Saqib et al. [24] used the state-of-the-art object detectors based on deep convolutional neural networks (R-CNN, Faster R-CNN, YOLO v2, and SDD) to detect human heads in natural scenes.

In [25], Peng et al. presented a method allowing detection of small heads. On the one hand, they exploited the multi-scale hierarchical functionalities created by deep convolutional neural networks, to obtain the most useful functionalities. And on the other hand, for the purpose of detecting small heads, they used the multi-scale cascade architecture. Likewise, Khan et al. [26] investigated the using of CNN’s multiscale representation fusion as a way to incorporate lower layers with upper layers for detection. This process was implemented in order to solve the problem of the non-detection of small heads.

Wang et al. [27] examined the pedestrian head detection problem by presenting a two-stage head detection framework; a fully convolutional network (FCN) to generate proposals followed by CNN to classify each proposal as head or background. Meanwhile, Khan et al. [28] proposed a framework for detecting human heads with a wide range of scale variance. The framework consisted of three scale-specific subnets with different RPNs that have been combined into a single backbone. They carried out a comparative study with the Faster-RCNN, SSD, YOLO, TinyFace and SD-CNN networks.

For the problem of crowded scenes, Stewart et al. [29] used a recurrent LSTM by taking an image as input and directly outputs a set of distinct detection hypotheses. They compared the loss functions for their application to choose the appropriate one. Furthermore, Vora et al. [30] presented an end-to-end head detection model using a single fully convolutional network. They focused on the effect of anchor size on system performance.

Skrabanek et al. [31] proposed a head detection system through a pipeline of vision passenger recognition systems based on ConvNets (using five different architectures) to ensure both feature extraction and classification. In addition, Khan et al. [32] presented a head detection deep model-based method. They took into consideration the scale variations of heads by generating scale aware proposals based on scale-map.

To solve the problem of low-quality images, Yudin et al. [33] focused on face and head detection, taking into account the noise of images captured in low light conditions. They used the FCN with clustering, the Faster R-CNN architecture, and the Mask R-CNN architecture. Otherwise, Le et al. [34] used Faster R-CNN and BaseNet as deep learning approaches to detect human heads. Their method automatically generated virtual shoulder annotations from the annotations of the head. Additionally, Chi et al. [35] developed head and human detection using deep learning networks by studying these two tasks in parallel to improve the efficiency of the results.

State-of-the-art research shows that, in this field, there are different applied deep learning approaches. Nevertheless, there is a gap in the application of the semantic segmentation by deep learning in this area.

## 3. Proposed System

The technique proposed is based on two parallel deeplabv3+ networks [8]; one detects the head masks (referred as system 1) and the other detects the centers of the heads (referred as system 2). This combination of the two networks gives the advantage of correcting the false detection of one by the other. The false detections that can be corrected are the heads over-detection and the overlapping of the heads, as you can see in Figure 1.

Figure 2 shows a synoptic diagram describing the steps of our proposed merged system for detecting human heads. We adopted our system by merging two DeepLabv3+ networks to obtain the semantic segmented images. The DeepLabv3+ network [8], which consists of an encoder and a decoder, is widely used for semantic segmentation. Three main steps are involved during the process in the DeepLab3+ architecture:Extract the features of the image with the backbone (DCNN) models. This serves as the first convolution tube to capture and hide maps of important features. We investigated the effect of modifying the base DCNN architecture on system performance, to be able to extract the optimal dense feature maps with higher semantic information.Control the size of the output feature maps by using Atreric convolutions in the last layers of the backbone. This is established using an atrous convolution.In the final stages, ASPP architectures classify the different pixels of the output image and process them through 1 × 1 convolutional layers to return to their original size.

After the segmentation process we applied the standard labelling of the related components technique [36] to detect all the presented objects.

## 4. Comparative Analysis and Evaluation

In this research on the one hand, a study of the effect of changing the basic DCNN architecture of Deeplabv3 + on detection performance is investigated, by comparing the results for each different architecture. On the other hand, and to show the superiority of our system, in contrast to system 1 and 2, a comparative study was made. In the same way, we have compared our proposed system with the SegNet [37] and U-Net [38]. The comparison was made according to the following criteria. All the scores are expressed as a percentage (%):(a)DataSet MetricsGlobal Accuracy gives us a quick and inexpensive estimate of the percentage of pixels correctly classified. The overall precision is therefore the ratio between the correctly classified pixels (for all the classes) and the total number of pixels.Mean Accuracy allows us to indicate the percentage of correctly identified pixels for all classes. it is used as well for evaluating the performance of the system during the head object detection. This score is used to calculate the proportion of correct predictions made by our model. It varies between 0 for an algorithm that never detects the right class, and 1.
(1)Accuracy=TP/(TP+FN)
where TP and FN represent respectively true positive and false negation errors.Mean IoU is the average IoU score of all classes in all images. It is used as well to indicates the ratio of correctly classified head pixels. IoU is used to establish a measure of statistical precision that penalizes false positives. IoU is the ratio of correctly classified pixels to the total number of ground truths and predicted pixels.
(2)IoU=TP/(TP+FP+FN)
where FP is false positive error.Mean BF Score is the average BF score of the overall images of that class. BF score is values in the range [0, 1]. A score of 1 means that the contours of objects in the corresponding class, in prediction and ground truth, are a perfect match.
(3)$$BF=\frac{TP}{TP+\frac{1}{2}\left(FP+FN\right)}$$Weighted IoU is used to reduce the impact of errors in small classes on the overall quality score (as well as our images have disproportionately sized classes). Average IoU of each class, weighted by the number of pixels in that class, is used as an evaluation matrix for this study.(b)Mean IoU by Bounding Box is referred to as bounding box ratio in this article. The goal here is to measure the object detection accuracy of bounding boxes detected with the labeled ground truth. For this purpose, we have transferred the detected objects (as masks) to bounding boxes, and calculated the percentage of overlap of the ground truth bounding boxes with the detected ones.(c)Difference of object counted is done by measuring the absolute value of the subtraction between the actual number of heads in each frame with the predicted one. For example, ±2 indicates that we have an under-detection or over-detection of two heads.

## 5. Results and Discussion

### 5.1. Datasets

#### 5.1.1. 1st Dataset

This dataset is prepared from scratch (acquisition to pre-processing and manual labeling of data), in which we have captured videos, from different locations with the RealSense D435 camera [39]. Out of these videos we obtained a sequence of images, from which the images were selected at random until a sufficient quantity was obtained. Only RGB images are used for the purposes of this dataset, which consists of 7000 images taken from seven locations. Figure 3 shows some examples taken from this dataset.

#### 5.1.2. 2nd Dataset

This dataset is applied in order to show the capacity of generalization of the proposed model by testing it directly on the models trained by the 1st dataset. That is to say that the system has the capacity to respond to new cases in even a completely new dataset. The second dataset contains 1000 images taken at an eighth locality, which differ from the 1st dataset locations. To further complicate matters we have changed the backgrounds and lighting conditions for this dataset. Figure 4 shows some examples taken from this dataset.

#### 5.1.3. 3rd Dataset

PAMELA UANDES dataset [9], which we used in this study, is composed of single-camera recording videos looking down on passengers alighting/boarding a metropolitan train (as it is shown in Figure 5). Ground truthed data of this dataset had been prepared manually by annotated people’s heads using 15 videos.

A preprocessing of this dataset is applied before working on it (we obtained 11,315 images at the end of this step). The following points were taken into account when preparing the images for this dataset:Cropping the original video to take only the interested region and to decrease the calculation time during the training of the models.The need to take consecutive images in pieces so that the tracking can be applied in future work.

As previously mentioned, the three datasets had been prepared in a specific way, in which two new ground truths were extracted from the original ground truth bounding boxes. The first by transforming the bounding boxes into ellipses, and the second by taking 10-pixels surrounding each center of objects. An example is illustrated in Figure 6.

Figure 7 shows the pixel frequency of the two classes (head and background) in dataset 1 and 3. To have good results, all the classes would have an equal number of observations. However, In our case, the classes are unbalanced which is a problem that will influence the performance of the proposed models. This imbalance can be detrimental to the learning process because learning is biased in favor of the dominant class, which is the background in our case. To resolve this problem, a class weight has been maintained.

### 5.2. Implementation Details

Our models were trained from scratch using the stochastic gradient descent (SGDM) optimizer with momentum of 0.9. 30 epochs were used, along with a mini-batch size of 8, learning rate to 0.001, and L2 regularization factor of 0.0005. All training algorithms were performed using MATLAB on a GPU of NVIDIA Quadro P5000 graphic card running on an operating system Windows.

Data augmentation was used for our application to improve network accuracy by randomly transforming the original data during training. For our application, random horizontal/vertical reflection, left/right random reflection, random X/Y translation of ±10 pixels, and random rotation were used.

Figure 8 shows the DataSet Metrics applied on the three datasets. The goal was to obtain the optimal base architecture DCNN to create the two parallel DeepLabv3+ networks. It can be observed that ResNet-50 achieved the best performance with a global accuracy of 98.58%, 96.61% and 99.14% for the three datasets respectively. That is explained by the fact that ResNet-50 has a greater number of parameters to be used, so it showed better performance as compared to the other ones. The Inceptionresnetv2 and the Xception architectures gave the lowest performance. This is because Inceptionresnetv2 focuses on computational cost, unlike ResNet, which focuses on computational accuracy. On the other hand, Xception is an extension of the Inception architecture which replaces the standard Inception modules with depthwise separable convolutions.

Figure 9 shows the head class metrics performance for the three datasets using our merged proposed system implemented on different architectures. We can see that head class Mean BF scores, using RasNet-50 architecture for the first and the third datasets, are 60.575%, 84.795% respectively. This indicated that the classes are balanced, even though they are not balanced in our datasets (as can be seen in Figure 7). This is due to the fact of applying the class weighting to balance the classes by calculating the median frequency class weights, and specifying them in the pixel classification layer of our network. For the second dataset we got a low BF score of 53.56%. This is because the network that was applied for this dataset is not custom-built for it, so it did not take its class frequencies into consideration.

We analyzed the performance of system 1, 2, and the merged one at different head count values on the three datasets. For this purpose, we measured the absolute value of the subtraction between the actual number of heads in each frame with the predicted one. For example, ±2 indicates that we have an over-detection or under-detection of two objects. The results are shown in Figure 10. Note that using ResNet-50, the proposed merged system was able to detect 95.12%, 61.60%, and 83.33% of the perfect object count, and 0.15%, 5.10%, and 1.67% in the under/over counting error, for the three datasets respectively.

Interestingly, we can see that our merged approach worked very well over a range of count values for the 3 different datasets. This confirms our intuition that the idea of training the system with two different ground truths can capture a very good head count.

The error in the head counts is due to the following causes: the lack of richness of the dataset, the quality of the images, the overlap of several heads, different size of the objects, and the single image processing. Besides, the counting of heads can be significantly improved by processing video in real time manner using multiple sequential image processing (future work).

Figure 11 shows the precision of main object detection of our proposed model compared to the ground truth object annotations of the three different datasets. For this purpose, Intersection over Union (IoU) was used as an amount to calculate the overlap between the bounding box of predicted truth and ground truth. Qualitatively, our approach yielded promising results as shown by the IoU Bounding Box. We can see that our proposed method using RasNet-50 gave us 86.55% IoU for our first dataset.

The objective thus was to study the impact of the basic architecture to create a DeepLabv3+ network by measuring the following characteristics of the only head class (which interests us the most): Accuracy, IoU and score BF. We used the boundary F1 (BF) score because it gives us a metric that tends to correlate better with human qualitative rating than the IoU metric, which is very important in this application. BF score here is more interesting than accuracy because the number of true negatives is not taken into account. And in imbalanced class situations we have real negatives that completely skew our perception of the performance of our system. A large number of true negatives will leave the BF score unmoved.

To show the performance of the proposal system, a comparative analysis was carried out on two deep learning semantic segmentation state-of-art models. Figure 12 shows the comparative results of our proposed system by comparing it with SegNet-VGG19 and U-Net. The proposed system showed that our approach achieved the best performance. Nevertheless, the overall outcome of the SegNet and U-Net results was most affected by noisy scatter points, especially around the edges. This can be observed by the granular segmentation at the boundary between the head objects and the background. Consequently, the datasets are not balanced, and with the propagation of images in the network through several alternative pooling layers, the resolution of the output feature maps is downsampled. This is also explained by the fact that SegNet and U-Net use fully convolutional neural networks during the preliminary image detection step. Here, masks and boundaries are put in place, and then the inputs are processed through a very deep network where the convolutions and accumulated groupings cause the image to decrease its resolution and quality. Hence, the results are a yield with a high loss of information. Our proposed model, however, took up the challenge by taking advantage of the Atrous Spatial Pyramid Pooling (ASPP) architectures.

In addition to the quality assessment, it was very interested to compare the algorithms in terms of processing time measured on the hardware infrastructure mentioned in Section 5.2. Table 1 summarizes the results.

For most semantic segmentation applications, we assume that a processing speed of less than 0.5 s is fast enough for image processing and less than 0.02 s is fast for video processing. From our previous results, we found that the model proposed with the ResNet-50 backbone architecture obtained the best result in terms of precision, IoU and BF score. With an average detection time of 211 ms, we can confirm that this choice is very suitable for this type of application using image processing. From Table 1, we can see that the model proposed with the Mobilenetv2 backbone is the only model that realizes the assumption of applying it for video processing with an average detection time of 18 ms.

Figure 12, Figure 13 and Figure 14 show experimental tests for the 1st, 2nd and 3rd datasets respectively. The selected test images are of low to high complexity. The first column represents images labelled in pixels (ground truth) by superimposing them on their raw images. The second, third, and fourth columns represent the predicted semantic segmentation of system 1, system 2, and our merged system, respectively. The fifth column represents a comparison of our system’s results with the expected ground truth. The colours highlight areas where the segmentation results differ from the expected ground truth (the green colour indicates positive false segmentation and magenta colour indicates true negative segmentation). And the sixth column represents the final output of the system with all detected objects in red bounding boxes.

The comparative results between the segmentation obtained by the considered methods and the desired ground truth for the three tested datasets, showed very satisfactory results of the proposed system compared to the others. From these experimental results, we can see the strong effect of the fusion of the two systems. Sometimes, the first system helps the second to correct its false detection and other times the reverse, which gives a major advantage for the proposed system. Experimental tests also showed that our proposed system detected almost all the heads correctly, even in a complex environment.

## 6. Conclusions

Robust and end-to-end segmentation solutions are very important to aid and improve many applications using human head detection. In this work, development of a fully automatic ensemble deep learning strategy which combines two parallel Deeplapv3+ networks to improve the performance has been established. A strategy based on not working directly with the bounding boxes given by the ground truth, but by two types of ground truths extracted from the bounding boxes is applied in this study (each is applied to one of the two parallel networks). An investigation, to have the optimal basic architecture to create the two parallel DeepLabv3+ networks, has been carried out in this article. Our approach has been implemented in our two private datasets as well as in a public dataset. The proposed system outperformed two deep learning semantic segmentation state-of-the-art methods: SegNet and U-Net, in which we achieved 99.14% in one of the tested datasets. The proposed system can also be applied in several semantic segmentation applications. Future work will be conducted on head object detection using semantic instance segmentation and head tracking using multiple sequential image processing. Future work will be carried out to apply and implement the proposed model on various datasets with multi-object classes.

## Figures and Tables

**Figure 1 sensors-21-05848-f001:**
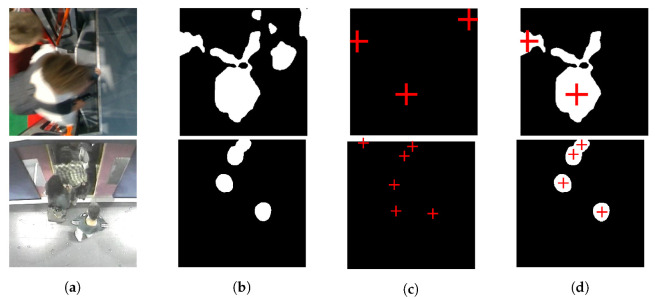
False detection that can be corrected by proposed system. (**a**) is raw image, (**b**,**c**) represent output of system 1 and 2. (**d**) is output of proposed system.

**Figure 2 sensors-21-05848-f002:**
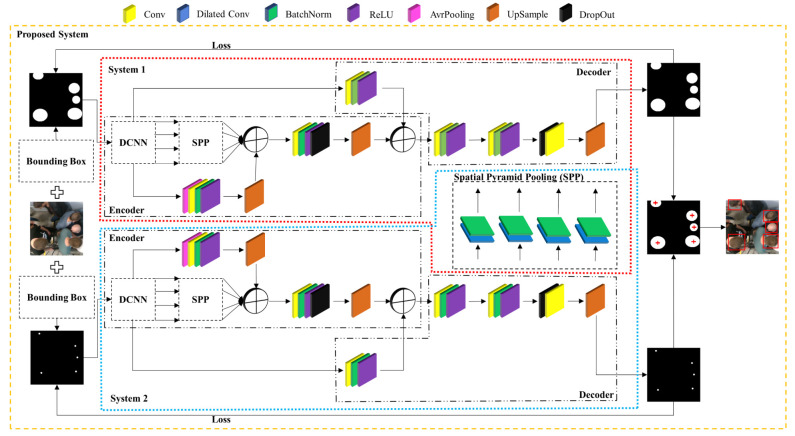
Block diagram describing the architecture of the proposed merged system for the detection of the human head.

**Figure 3 sensors-21-05848-f003:**
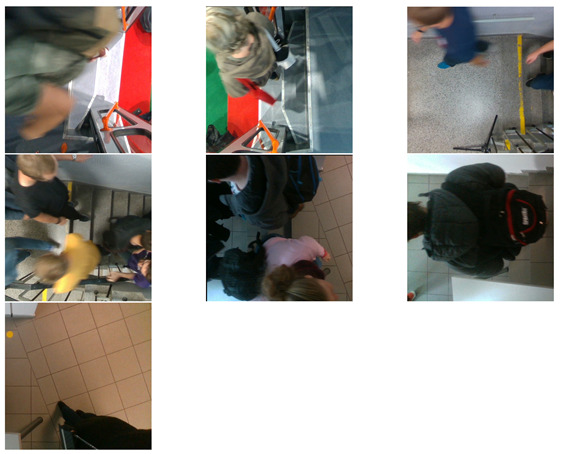
Some examples taken from the first dataset.

**Figure 4 sensors-21-05848-f004:**
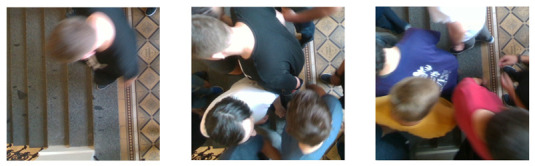
Some examples taken from the second dataset.

**Figure 5 sensors-21-05848-f005:**
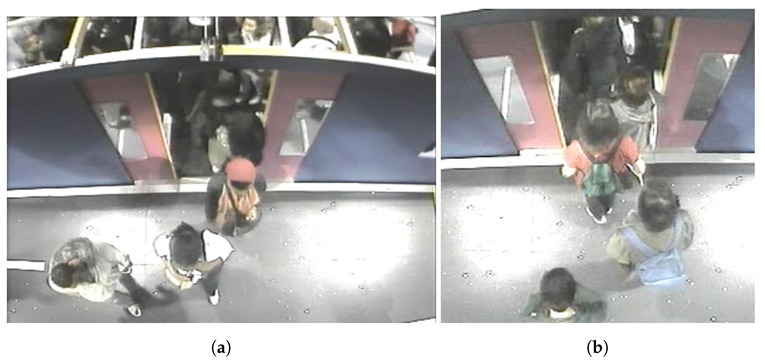
An example taken from the third dataset in which (**a**,**b**) represent respectively before and after the preprocessing of the dataset.

**Figure 6 sensors-21-05848-f006:**
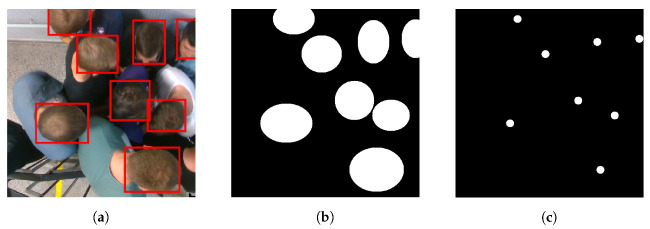
The preparation of the ground truth. (**a**–**c**) respectively represent an example of: original bounding boxes superimposed on its raw image, the ground truth of system 1, and the ground truth of system 2.

**Figure 7 sensors-21-05848-f007:**
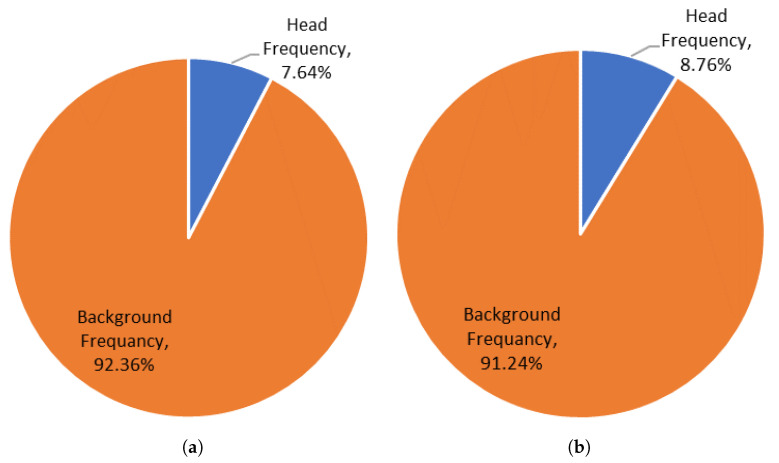
Pixel frequency of both classes (head and background) in dataset 1 (**a**) and 3 (**b**).

**Figure 8 sensors-21-05848-f008:**
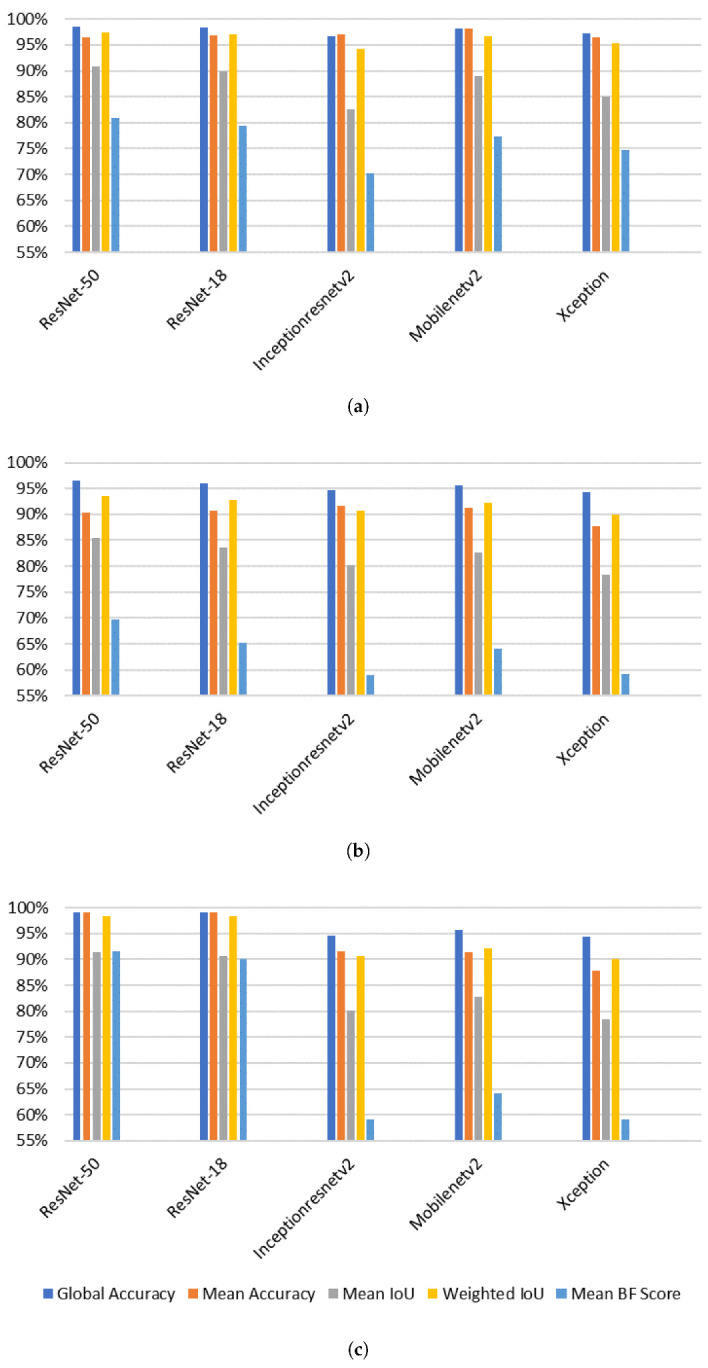
DataSet Metrics applied to the three datasets referenced respectively by subfigures (**a**–**c**).

**Figure 9 sensors-21-05848-f009:**
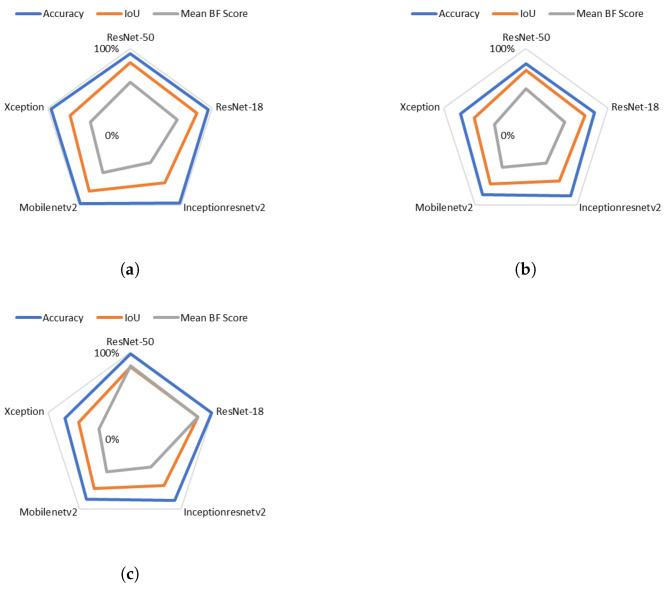
The impact of the DeepLabv3+ base architecture on the performance of the head class. (**a**–**c**) represent the results of the three datasets.

**Figure 10 sensors-21-05848-f010:**
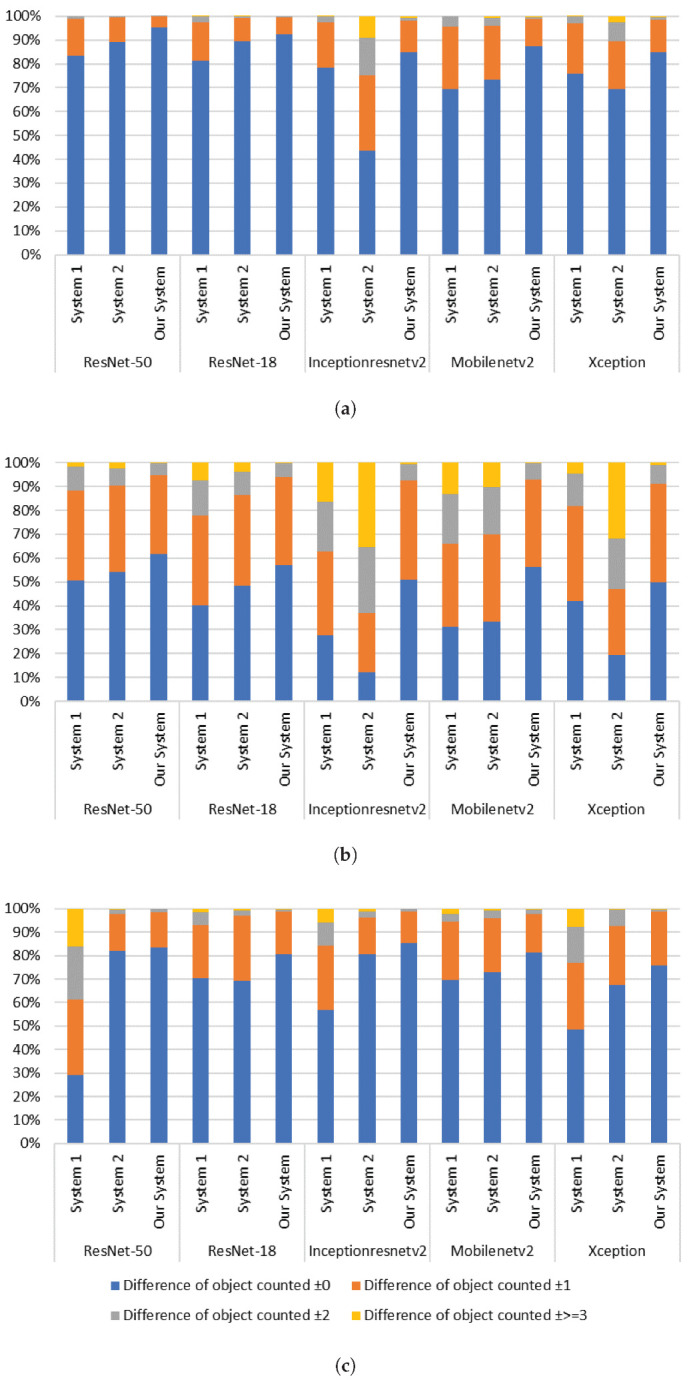
Performances of system 1, 2 and of the merged system for head counting, (**a**–**c**) respectively represent the results of the 1st, 2nd and 3rd dataset.

**Figure 11 sensors-21-05848-f011:**
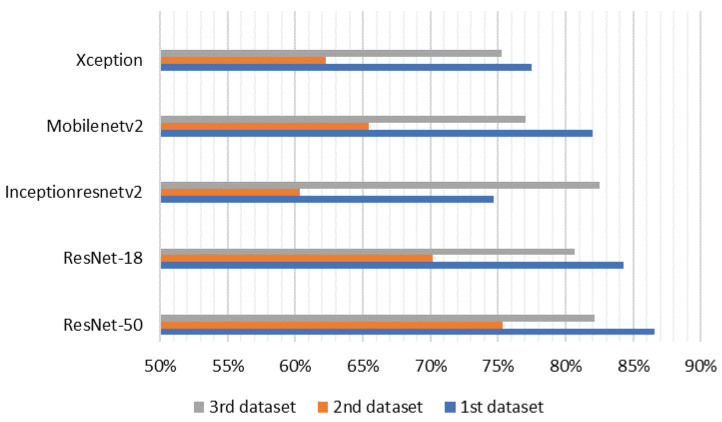
Average bounding box IoU for the three datasets tested.

**Figure 12 sensors-21-05848-f012:**
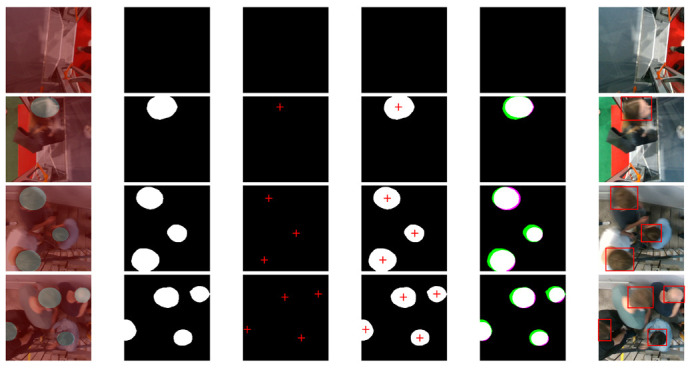
Experimental results on the first dataset. The first column represents images labelled in pixels (ground truth) by superimposing them on their original images. The second, third, and fourth columns represent the predicted semantic segmentation of system 1, system 2, and our merged system, respectively. The fifth column represents a comparison of our system’s results with the expected ground truth. The sixth column represents the final output of the system with all detected objects in red bounding boxes.

**Figure 13 sensors-21-05848-f013:**
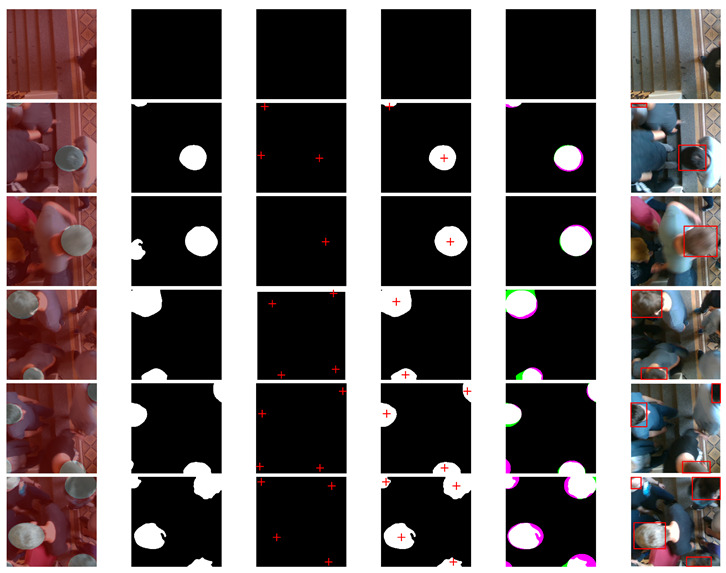
Experimental results on the second dataset. The first column represents images labelled in pixels (ground truth) by superimposing them on their original images. The second, third, and fourth columns represent the predicted semantic segmentation of system 1, system 2, and our merged system, respectively. The fifth column represents a comparison of our system’s results with the expected ground truth. The sixth column represents the final output of the system with all detected objects in red bounding boxes.

**Figure 14 sensors-21-05848-f014:**
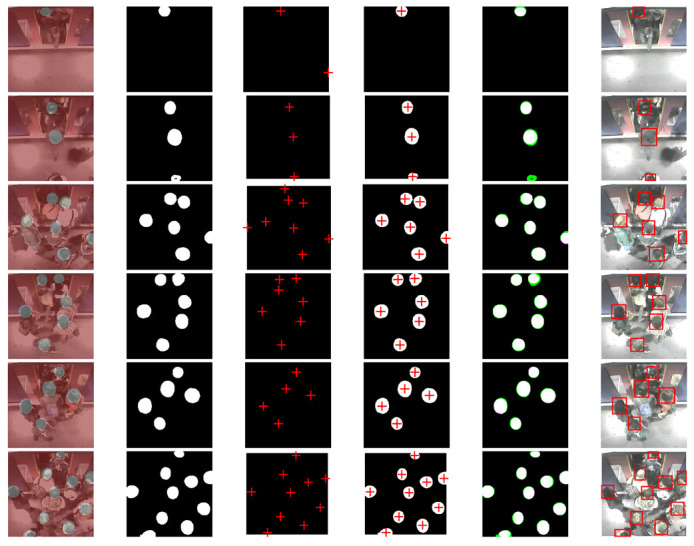
Experimental results on the third dataset. The first column represents images labelled in pixels (ground truth) by superimposing them on their original images. The second, third, and fourth columns represent the predicted semantic segmentation of system 1, system 2, and our merged system, respectively. The fifth column represents a comparison of our system’s results with the expected ground truth. The sixth column represents the final output of the system with all detected objects in red bounding boxes.

**Table 1 sensors-21-05848-t001:** The computational cost of the tested models.

Algorithms	Parameters (Million)	Average Training Time (Hour:Minute:Second)	Average Detection Time per Image (Millisecond)
Proposed model with Rasnet50 backbone	∼46	35:50:24	211
Proposed model with Rasnet18	∼22	17:02:12	101
Proposed model with Inceptionresnetv2	∼112	87:14:47	513
Proposed model with Mobilenetv2	∼4	3:06:57	18
Proposed model with Xception	∼46	36:04:35	212
Unet	∼30	23:22:10	137
Segnet	∼30	23:46:04	138

## Data Availability

The data presented in this study are available in http://velastin.dynu.com/PAMELA-UANDES/whole_data.html.

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
