# Peer review of "New End-to-End Strategy Based on DeepLabv3+ Semantic Segmentation for Human Head Detection"

_sensors, 2021, doi:10.3390/s21175848_

Round 1

Reviewer 1 Report

This paper proposed a new end-to-end strategy based on DeepLabv3+ semantic segmentation to improve the accuracy of head detection.

Pros:

1. Taking a novel end-to-end strategy based on DeepLabv3+ semantic segmentation into account is an effective way to improve the accuracy of head detection.

2. In experiment part, authors use a number of data sets to conduct experiments to prove the performance of the proposed model, which is worthy to be mentioned.

Cons:

1. In METHODOLOGY, The models and methods proposed in this paper are based on two parallel Deeplapv3+. This is not a substantial innovation.

2. This paper developed a new approach based on two parallel Deeplapv3+ to improve the performance of the person detection system. In the experiment part, the speed of head detection should be discussed based on two parallel Deeplapv3+.

To sum up, I would say this paper is not publishable for Sensors so far.

Author Response

Dear Reviewer,

Our reply is attached as Pdf file.

Best regards

Reviewer 2 Report

This paper leads to segment and detect human head with ensemble deep learning in which consist of two parallel Deeplapv3+ networks
+good comparison by applying different architecture such as ResNet-50, ResNet-18, Mobilenetv2, Xception
+introducing two private databases in which it contains 1000 images taken at an eighth locality
+good literature review 

Author Response

Dear Reviewer,

We thank the reviewer for their positive feedback.

Best regards

Round 2

Reviewer 1 Report

The manuscript has been sufficiently improved and can be accepted in its present form for publication.